# Qualitative interview study exploring the perspectives of pregnant women on participating in controlled human infection research in the UK

Robert B Dorey [ID],[1,2] Anastasia A Theodosiou,[1,2] Robert C Read [ID],[1,2] Tushna Vandrevala,[3] Christine E Jones[1,2]

[1]Clinical and Experimental Sciences, Faculty of Medicine, University of Southampton, Southampton, UK
[2]NIHR Southampton Clinical Research Facility and NIHR Southampton Biomedical Research Centre, University Hospital Southampton NHS Foundation Trust, Southampton, UK
[3]Centre for Applied Health and Social Care Research, Kingston University and St George's University of London, London, UK

**Correspondence to**
Dr Robert B Dorey;
rob.dorey@ggc.scot.nhs.uk

## ABSTRACT

**Introduction** Pregnant women have been historically excluded from interventional research. While recent efforts have been made to improve their involvement, there remains a disparity in the evidence base for treatments available to pregnant women compared with the non-pregnant population. A significant barrier to the enrolment of pregnant women within research is risk perception and a poor understanding of decision-making in this population.

**Objective** Assess the risk perception and influences on decision-making in pregnant women, when considering whether to enrol in a hypothetical interventional research study.

**Design** Semistructured interviews were undertaken, and thematic analysis was undertaken of participant responses.

**Participants** Twelve pregnant women were enrolled from an antenatal outpatient clinic.

**Results** Participants were unanimously positive about enrolling in the proposed hypothetical interventional study. Risk perception was influenced by potential risks to their fetus and their previous experiences of healthcare and research. Participants found the uncertainty in quantifying risk for new research interventions challenging. They were motivated to enrol in research by altruism and found less invasive research interventions more tolerable.

**Conclusion** It is vital to understand how pregnant women balance the perceived risks and benefits of interventional research. This may help clinicians and scientists better communicate risk to pregnant women and address the ongoing under-representation of pregnant women in interventional research.

## STRENGTHS AND LIMITATIONS OF THIS STUDY

⇒ Relatively small sample size, although semistructured interview format allowed for rich data to be gleaned and provided detailed themes, which were examined.
⇒ Conducted with the group of interest, engaging individuals for whom pregnancy-related research is most important.
⇒ Participants had gained experience throughout pregnancy in weighing decisions and balancing medical decisions, allowing for excellent engagement in discussion about study enrolment.
⇒ Participants had relatively high educational attainment and so may not be representative of the broader population.

restricted following a number of highly publicised cases of harm caused by medical interventions in pregnancy.[1] For instance, the use of thalidomide in pregnancy resulted in many thousands of congenital anomalies and infant deaths in the 1960s, while antenatal use of diethylstilbestrol was recognised as a cause of cancer in women and their female offspring in the 1970s.[1–3] Due to perceptions of greater susceptibility to harm, pregnant women were increasingly excluded from interventional research and even reclassified as a 'vulnerable population' by the US Department of Health and Human Services in 1975.[1 2]

The 1980s saw a shift towards greater inclusion of pregnant women in research studies. With dramatically increasing medical costs in the USA, research offered the public the possibility of subsidised medical care.[1] Furthermore, before antiretroviral therapies were made widely available for the treatment of HIV, clinical research participation was viewed as a means of accessing medication for an otherwise fatal infection. As the incidence of HIV in women increased, there was a rise in public opinion that the exclusion

## BACKGROUND

### Interventional research in pregnant women

Interventional research in pregnancy has remained contentious since the formalisation of human research ethics in the mid-20th century.[1 2] Although neither the Nuremberg Code (1947) nor the World Medical Association's Declaration of Helsinki (1964) explicitly referred to pregnant women, their participation in interventional research was

of women from HIV research studies was discriminatory. This context increased political pressure to expand inclusion of pregnant women in interventional research.[1] Subsequently, in 1985, the US Public Health Service Task Force publicly recognised a lack of data specific to women's health, and Congress established a Department of Women's Health at the National Institutes for Health through the passage of the Women's Health Equality Act (1990), thus mandating research in this area.[2] This new sentiment was further enforced by international guidance from Council for International Organizations of Medical Sciences in 1993 characterising the exclusion of pregnant women from research as 'unjust'.[4]

Despite this progress, concerns remain among clinicians and researchers that interventional studies may pose a risk of serious harm to the mother and fetus. There is evidence that clinicians overestimate the risk of teratogenicity caused by medications during pregnancy.[5] Further, ethics committees appear more likely to perceive interventional studies in pregnancy as high risk, due to difficulties in quantifying or excluding risk of harm to the fetus.[6] Indeed, a study involving interviews with ethics committee members found that many members would not question the exclusion of pregnant women from research, and felt it was easier to exclude them from interventional research and opt for observational studies instead.[6] These concerns may contribute to ongoing exclusion of pregnant women from interventional research. Indeed, of 172 medications approved by the US Food and Drug Administration between 2000 and 2010 and 97.7% has 'undetermined' teratogenicity risk, while safety data relating to pregnancy was reported as 'none' for 73.3%.[7] Even within the last decade, pregnant women were excluded from trials of Ebola treatment and vaccines during the outbreak of 2013–2016,[8] and from up to 80% of relevant interventional studies during the first year of the SARS-CoV2 pandemic in 2020, often without specific pregnancy-related safety concerns regarding the investigational medicinal product.[9] In the absence of interventional research, many treatment decisions in pregnancy, therefore, rely on extrapolating from research in non-pregnant individuals and postmarketing data, which may be associated with the use of outdated or inferior treatments and even worse clinical outcomes.[3 10]

Even when pregnant women are eligible to participate in interventional research, they may be reluctant to enrol. Concerns among pregnant women include the harm that enrolling in interventional studies may cause their fetus, and a lack of knowledge about the investigational product.[11 12] Furthermore, interview studies have highlighted a perception that enrolling in research may cause participants stress, something they wish to minimise during pregnancy.[12 13] This may be a legacy of the exclusion from research that pregnant women have experienced, and continues to contribute to the perception that all research in pregnancy is potentially harmful, regardless of the relative risks associated with a particular intervention.[11] An improved understanding of how pregnant women balance risks and make decisions regarding research involvement may allow better engagement of pregnant women by research staff and allow individuals to overcome perceived barriers.

## Controlled human infection research in pregnancy

Controlled human infection (also known as human challenge) is a type of interventional research that involves exposing participants to a defined number of live microorganisms to investigate host response, colonisation and infection kinetics, and even the effect of antimicrobial products or vaccines. Much like research in pregnancy, this type of research presents unique ethical and practical considerations over and above those shared by all interventional research.[14 15] These include balancing the risk to the participant and their contacts by exposure to a potentially harmful organism, against the direct benefits to the participant and potential future health benefits for society.

Prior to the study described below, ethically approved controlled human infection research had not been conducted in pregnant women. A growing body of evidence has highlighted the importance of maternal to infant microbe transmission in shaping the neonatal upper respiratory microbiome and long-term adverse health outcomes such as respiratory infections and asthma.[16 17] These findings have led to calls for interventional research in pregnancy and early life, to further investigate the role of maternal–infant microbe transmission in child health, and to explore new avenues for preventing and treating childhood infections, such as mucosal inoculation with beneficial microbes.[18] Thus, the Lactamica 9 study was proposed as a human challenge study in pregnancy, using *Neisseria lactamica* nasal inoculation in pregnant women to investigate for mother-to-infant upper respiratory commensal transmission and impact on infant microbiome and immune development.[16]

*N. lactamica* is a type of harmless commensal bacteria that has been well characterised in human challenge research involving healthy non-pregnant adults. It is found in the upper respiratory tract of over 40% of infants aged 1–2 years, although it is present in fewer than 10% of adults and neonates.[19 20] *N. lactamica* exhibits an inverse relationship with *Neisseria meningitidis* colonisation and invasive disease, and in controlled human infection studies, it has been shown to displace *N. meningitidis* from the upper respiratory tract by inducing cross-reactive humoral and cellular immunity.[21–23] In the immunocompetent host, *N. lactamica* does not cause clinical disease and has been safely administered intranasally to over 400 non-pregnant adults in human challenge studies.

Given the novelty of the proposed Lactamica 9 study, and the unique ethical and practical considerations of performing human challenge research in pregnant women, a preparatory study was conducted using semistructured interviews with pregnant women. These interviews aimed to gain insights into pregnant women's perception of risk and decision-making when

considering hypothetical participation in a controlled human infection study. The results of the interview study are presented below and were used to inform the design and ultimate execution of the Lactamica 9 study. In doing so, we aimed to identify and address barriers to research involvement, facilitate better engagement with pregnant women and ultimately improve equitable participation in this population.

## METHODS

Women were approached in an antenatal clinic at a large teaching hospital in the UK, between March 2019 and February 2020. All women attending an antenatal clinic were approached while in the waiting room, and their initial interest was determined. Those who expressed interest were given more information about the study, and, if they wished to proceed, informed consent was gained. Women were eligible if they were aged over 18 years and receiving antenatal care at the site; and were excluded if they were not able to understand written and spoken English.

Face-to-face, semistructured interviews were carried out with 12 pregnant women, lasting between 30 and 60 min each, and all interviews were audiorecorded. The interview guide consisted of open-ended questions and prompts, and follow-up questions were asked to further explore themes raised by participants (see online supplemental material 1). The interview asked participants to reflect on their hypothetical participation in a proposed study and explored participants' understanding of invasive meningococcal disease, perception of risk and acceptability of different study designs, procedures and interventions. In particular, participants were asked to consider live bacterial inoculation using nasal drops, and collection of upper respiratory (nose, throat and mouth) swabs, breast milk and blood from themselves and their newborn baby. Following the interviews, recordings were transcribed verbatim and transcripts were anonymised. No further interviews were conducted once data saturation (the point at which each new interview produces only previously discovered data) was reached.

Data were managed and analysed by using NVivo (V.1.6.1). Data analysis was performed in five stages as described by Braun and Clarke[24]: (1) familiarisation—researchers were familiarised with data prior to analysis so that broad content was understood; (2) code generation—transcripts were summarised using codes, where each code described a segment of data using a key analytic idea; (3) searching for themes—following code generation, researchers combined different codes under broader generalised patterns, called themes, which described commonalities between codes; (4) reviewing themes— once identified, codes were better defined using central organising concepts, and these themes were reviewed to ensure that they accurately represented the data, particularly as it related to the research theme and (5) defining and naming themes. Codes and themes were cross-checked, discussed and reviewed with two other researchers (CEJ and TV) to ensure accurate representation of participant responses. Themes were analysed in relation to the research question, and data were interpreted in this context (online supplemental material 2).

### Patient and public involvement

Prior to this study, pregnant women were approached while attending an antenatal parent evening at a maternity hospital. Pregnant women and their partners were engaged in an informal discussion regarding the proposed study. The study design was explained, including interview content, duration and study aims. Opinions were then gathered regarding the phrasing of questions, possible locations of recruitment, and strategies and factors that would encourage or dissuade engagement. Eight couples were involved in this process, and this information informed the design of the presented study in a number of ways. First, participants were recruited at an antenatal clinic in a maternity hospital as it was felt that this would reduce disruption for participants enrolling in the study. Second, the language used in interviews was changed so that it might be less alarming to participants; for example, the scientific term 'bacteria' was preferable to colloquial term 'bugs'. Finally, it was felt that an interview duration of 30–60 min would be acceptable, especially given intervals of waiting during the antenatal clinic from which study participants were recruited.

## RESULTS

Twelve pregnant women were interviewed regarding their perceptions of risk and their decision-making about hypothetical participation in a controlled human infection study in pregnancy. Participants were women of childbearing age (25–40 years) with a range of educational attainment (see table 1). Broadly, participants were white British (11/12), and seven participants did not have any children. The majority of participants (11/12) would, in principle, agree to participate in the interventional study proposed.

Overall, our data suggest that pregnant women have a heightened perception of risk, compared with when they were not pregnant. They balanced the risks and benefits of enrolling in an interventional study in the context of their individual experiences and were motivated by the potential for developing new treatments. The acceptability of a research study was closely linked to the interventions that would be undertaken, and participants perceived uncertainty about risk to be equivalent to a high risk of harm (see table 2 for themes and subthemes).

### Theme 1: heightened perception of risk during pregnancy

Participants were concerned about the risks associated with participating in research during pregnancy. This concern was attributed to two main factors: concern for the health of their unborn baby, and that they themselves may be at higher risk by virtue of being pregnant. Women

**Table 1** Demographic characteristics of participants. GCSE/BTEC: state examinations for 16 year olds. AS/A-levels or Scottish higher: state examinations for students aged 17-18 years. BSc/BA: Bachelor of Science/Bachelor of Arts degree (undergraduate degrees).

| Characteristic | Participant no (%) |
|---|---|
| Marital status | |
| Single | 7 (58) |
| Married | 5 (42) |
| Gestation | |
| First trimester | 1 (8) |
| Second trimester | 6 (50) |
| Third trimester | 5 (42) |
| Gravida | |
| Gravida 1 | 6 (50) |
| Gravida 2 | 5 (42) |
| Gravida 4 | 1 (8) |
| Parity | |
| Parity 0 | 7 (58) |
| Parity 1 | 4 (33) |
| Parity 3 | 1 (8) |
| Ethnicity | |
| White British | 11 (92) |
| Any other Asian background | 1 (8) |
| Time in the UK | |
| Since birth | 11 (92) |
| 15 years or more | 1 (8) |
| Disability or long-term health condition | |
| No | 9 (75) |
| Yes | 3 (25) |
| Highest qualification | |
| GCSE/BTEC or equivalent | 3 (25) |
| AS/A-levels or Scottish higher or equivalent | 3 (25) |
| BSc/BA or equivalent | 5 (42) |
| PhD or equivalent | 1 (8) |
| Previous participation in research | |
| No | 11 (92) |
| Yes | 1 (8) |

expressed a heightened perception of risk while pregnant, with even a small amount of risk to their baby being intolerable, although a degree of risk to themselves was felt to be more acceptable.

> My first and foremost concern would obviously be the safety to my baby. And then, secondary to that, myself (Participant 9, 26 weeks gestation, second pregnancy)

> With my baby, even a small amount of risk is not tolerable (Participant 1, 28 weeks gestation, first pregnancy)

**Table 2** Summary of themes and subthemes present in interviews with pregnant women

| Theme | Subtheme |
|---|---|
| Heightened perception of risk during pregnancy | Gestational age changes risk perception |
| | Babies cannot consent |
| Balancing risks and benefits by pregnant women | Prior understanding and experiences influence decision-making |
| | Conflation of uncertainty and risk |
| Factors which encourage participation in research | Altruism |
| | Research processes are reassuring |
| | Acceptability of study procedures |

### Subtheme 1.1: gestational age changes risk perception

Women's perception of risk was related to the gestational timing of the proposed research study, although this effect varied between individuals. Some participants reported that they would avoid participating in research in early gestation, as they perceived a greater risk of complications and felt more uncertain about their pregnancy at this time. Conversely, other participants reported greater reservations about participating later in pregnancy, as they expressed more attachment to their fetus, greater confidence that the pregnancy was going to progress to birth, and that their fetus was more like an individual (in part because they could feel it move).

> I suppose at 36 weeks … If anything were to go wrong, 36 weeks is a very good threshold that they can convert to be induced I suppose. But yeah, that has made me think more about participating (Participant 7, 26 weeks gestation, first pregnancy)

> If I knew there was going to be a high risk to harming the baby, then I probably wouldn't because if you're doing it later on in pregnancy. Obviously that baby is already a part of you. You can feel it move, you can feel it kick and you get used to its routine and stuff like that. (Participant 4, 34 weeks gestation, first pregnancy)

### Subtheme 1.2: babies cannot consent

For some participants, the inability of the fetus to consent to research participation was paramount when considering hypothetical involvement in the proposed study. Some suggested that, since the research could theoretically cause long-term complications, they did not feel it was acceptable to consent on the unborn child's behalf. Furthermore, the baby's inability to consent would increase their own feelings of guilt if complications or adverse outcomes arose.

> I go into it knowing about it, my baby doesn't. So, I would be more worried about what it would do to my baby. (Participant 2, 12 weeks gestation, first pregnancy)

## Theme 2: balancing risks and benefits by pregnant women

For most participants, their views on hypothetical participation in the proposed human challenge study were based on a balance of perceived benefits of enrolment against potential risks and adverse events. Being pregnant brought with it, an even greater responsibility to carefully balance these risks and benefits.

### Subtheme 2.1: prior understanding and experiences influence decision-making

Previous experiences of invasive meningococcal disease influenced how they viewed their hypothetical involvement in such research. Women with personal experience, or who knew someone affected by the condition being investigated, gave greater weight to the importance and potential benefits of the study and were, therefore, generally supportive of study participation.

> It wasn't on my radar, until a friend of mine has recently had a scare with meningitis. Her little girl had a rash that wouldn't disappear, so they took her up to hospital. But it seemed it wasn't meningitis. Then, they kind of, it does play on your mind a bit. (Participant 7, 26 weeks gestation, first pregnancy)

Previous experience of research or medical care also weighed greatly on participants' decision-making. More specifically, traumatic experiences, particularly related to medical procedures, increased perceived risk. For example, a previous child having a difficult experience with venesection would make a study requiring neonatal blood tests appear more risky or unacceptable.

> I had to take my daughter quite young for a blood test. And I remember her, she was really distressed (Participant 10, 33 weeks gestation, second pregnancy)

For some participants, risk perceptions about study participation during pregnancy appeared entrenched and binary; either research would provide significant benefits warranting participation, or posed significant risks precluding participation. For these individuals, perspectives on risks and benefits of research did not change during the interview, despite information and safety data provided by the research team. Further, their stance did not appear specific to this particular study, but rather applied more broadly to research participation regardless of proposed interventions or procedures. In contrast, other individuals who were undecided about hypothetical study participation at the start of the interview appeared to favour participation in the hypothetical study, following provision of further information by the interviewer.

> Yeah, but if you already knew, I take it as if you already know information anyway and you have an opinion about it, you're going to go for it or against it. Either way. Your mind is already made up … I think it would just reassure the ones that maybe are going to, maybe people sitting on the fence might be swayed a bit more, but the "no"s definitely wouldn't be, I don't think it would affect, I don't think it would change their mind. (Participant 6, 27 weeks gestation, first pregnancy)

### Subtheme 2.2: conflation of uncertainty and risk

Participants were wary of the uncertainties involved in enrolling in an interventional study, especially one that has not previously been undertaken in pregnancy. They expressed difficulty in balancing proposed benefits with the potential for unknown complications, and considered the impact on their fetus more than themselves. They perceived uncertainty about potential harm in much the same way as they would know risks. Indeed, they struggled to rationalise participation when research staff could not unequivocally guarantee no risk associated with study participation.

> Because, still, you can't guarantee it won't cause any problems (Participant 8, 34 weeks gestation, second pregnancy)

Specific concerns were that the intervention itself could cause harm. While participants recognised that undertaking research is required to improve care for pregnant women and neonates, some felt that participation in interventional research posed greater risk to them personally than the prospect of insufficient research. This perception was heightened as they were told there may be no direct benefits to the mother or neonate of participating.

> I don't know. I really don't know. I think that sitting on the fence. You would never want to see your baby ill, but you also wouldn't want to put your baby in jeopardy if this is something that didn't work… That's just the way I think about it. (Participant 1, 28 weeks gestation, first pregnancy)

## Theme 3: factors which encourage participation in research

Despite the perceived uncertainties and risks, the participants were nearly unanimous in support of hypothetical participation in such a human challenge research study. Their support was mainly related to specific features of the proposed study, as well as a desire to further the care of pregnant women and neonates in general.

### Subtheme 3.1: altruism

A significant motivator for engaging with research was altruism. There was a perception that engaging with research as a pregnant woman could improve the care available to pregnant women and neonates, providing benefits to both themselves and others. Participants identified with other pregnant women and new parents, and felt motivated to improve research to reduce the chance of other families bearing the burden of disease and illness.

> I guess helping in medical research as well, meningitis can be fatal. And if you can stop another child and

another family having to go through that, that would be something nice to do. (Participant 4, 34 weeks gestation, first pregnancy)

I just think, why wouldn't you try and help if you could? (Participant 6, 27 weeks gestation, first pregnancy)

### Subtheme 3.2: research processes are reassuring
Participants were reassured by the safety procedures in place regulating clinical trial. They felt confident that inclusion and exclusion criteria meant they would not be enrolled if it were inappropriate or unsafe. Further, they took comfort in knowing that they, and their baby, would be regularly reviewed by a clinician; and that if adverse events arose (affecting other participants or themselves), they would be investigated and managed according to study procedures.

And knowing all of these checks, there are going to be hurdles you have to get through to take part, is good. (Participant 2, 12 weeks gestation, first pregnancy)

I would feel happier about it now knowing there are contingencies in place. (Participant 2, 12 weeks gestation, first pregnancy)

### Subtheme 3.3: acceptability of study procedures
The proposed study intervention and sampling procedures influenced perceived acceptability. When presented with a range of sampling procedures, participants expressed greater willingness to participate if less invasive procedures were proposed and even appeared to perceive reduced risk regarding the study as a whole (despite not altering the proposed study intervention). In particular, participants felt that research involving needles was less acceptable than those without needles, and they described a preference for heel prick sampling over venesection to acquire neonatal blood. Moreover, the proposed intervention (inoculation with live *N. lactamica*) was considered more natural, more acceptable and less risky, if administered via nasal drops than via intramuscular injection.

That would be more acceptable to me, just because, it's (heel prick blood test) quick, it's simple, and I think it's not as, I think it's not as dramatic as a baby, for a baby, but yeah, it's easier (Participant 6, 27 weeks gestation, first pregnancy)

I think that's fine, it's not that invasive, to you or the baby, it's just a spray. So that part wouldn't really put me off. (Participant 3, 26 weeks gestation, second pregnancy)

Well, I guess the natural version will always be more preferable (Participant 6, 27 weeks gestation, first pregnancy)

## DISCUSSION
Understanding and communicating risk is complex, and most research to date on risk perception in pregnant women has focused on lifestyle measures rather than participation in interventional research.[25–28] This interview-based study identified differences in participants' conceptualisation of risk, although most participants cited uncertainty, invasiveness or personal experience as contributors to their perception of risk. Further, many participants perceived risk as binary, rather than graduated based on exposure and study-specific considerations.

### Uncertainty affects perception of risk
The challenge of understanding and communicating risk may be compounded by pregnant women's apparent conflation of risk and uncertainty. In the context of research participation, risk refers to the probability of harm following an intervention, while uncertainty relates to unknown or unexpected consequences (which may or may not be harmful). Participants often equated uncertainty with harm, viewing it as a major barrier to hypothetical participation in interventional research. This finding has been reported in past literature, with pregnant women reporting a preference to let matters take their 'natural course'.[12 13] Thus, they distinguish between an adverse outcome arising through inaction (ie, choosing not to participate in research) and through an action that subsequently causes harm (ie, agreeing to a research intervention).[13]

This conflation of risk and uncertainty may be a reflection of modern obstetrics, where, due to advances in antenatal care, so much is known about a pregnancy that uncertainty is marginalised and so the unknown becomes perceived as a risk.[29 30] This misconception is pervasive and affects not only how pregnant women perceive themselves, but also how healthcare staff and researchers see them, and may thus contribute to the ongoing exclusion of pregnant women from interventional research.

### Invasiveness of intervention affects perception of risk
The nature of research interventions and procedures influence how risk is perceived by potential participants. Based on our small sample of pregnant women, participants expressed a preference for nasal compared with intramuscular inoculation and for heel-prick sampling over infant venesection. Past research has reported a reduced likelihood of study enrolment if venesection is required in children,[31] while painful interventions in general are associated with negative participant experience while enrolled in research studies.[32] The reasons for aversion to needles are likely complex; however, in our participants, procedures involving needles were perceived to be more invasive, and therefore less acceptable and more risky. Furthermore, our participants felt that the proposed intervention utilised a natural process (neonatal colonisation with maternal bacterial flora), and they found this more acceptable than other interventions they were asked to reflect on, such as injectable vaccines. The preference of society for natural products is well described,[33 34] and it may be that using non-needle-based

Recommendations for research

- Redress the recruitment of pregnant women in research. Recent examples have shown that conducting research in pregnancy can be made safe and transparent. Furthermore, our study demonstrates that some pregnant women may be enthusiastic about engaging with research studies.

- Ensure that research is acceptable and feasible to pregnant women. Pregnant women should be engaged early in the design of research, through Patient and Public Involvement work, and detailed discussions held about study requirements, particularly the utility and scientific justification of each of these.

- Our data suggest that the method of intervention administration matters greatly to participants. Researchers should consider whether alternative methods of administration may better engage pregnant women in research.

- Recognise that the exclusion of pregnant women from research has a demonstrable impact on their medical care, and may in fact pose greater risk to them than study enrolment.

**Figure 1** Recommendations for research. Four recommendations for how to improve equity of research for pregnant women.

interventions may allay concerns and alter risk perception of interventional research in pregnancy.

### Prior health-related experience affects perception of risk

The interviews demonstrated that how participants perceive risk and weigh decisions is greatly influenced by their prior experiences. The effect of previous experience on risk perception has previously been described and is maintained even if there is no effect on absolute risk; for example, in genetic diseases affecting non-familial relations.[35 36] Research participants who decline to be involved in studies have reported different levels of perceived risk to those who go on to enrol in research, and the information presented does not appear to change the likelihood of enrolment.[11] It appears that prior life experience is important in risk perception and decision-making, and even before participants are approached, they may be predetermined to either enrol or not based on these experiences.

Previous studies have demonstrated that, while clinicians typically present risk numerically (in terms of incidence or odds), pregnant women perceive risk in a more individualised way, relying more on personal experience than epidemiology.[37]

### Making research and treatment more equitable

There has been a shift away from protectionist policies in recent years, and inclusion of pregnant women within research has improved.[2] However, there remains a wide disparity between engagement of pregnant women compared with non-pregnant participants in interventional research.[8 9] Even recent calls 'to protect (pregnant) groups through research, not just from research'[38] retain vestiges of protectionist sentiment: the focus remains on researchers and governing bodies assessing risk on behalf of pregnant women, rather than empowering women to assess risk and decide whether to enrol.

### Communicating risk

We have highlighted, as have others,[37 39] that understanding risk is challenging for all involved in research. Research needs to be made more accessible to pregnant women by improving the appreciation of the differences in risk conceptualisation between researchers and pregnant women, and improved understanding that balancing risk may depend on personal previous experience.

### Patient and public involvement

A major barrier for pregnant women considering whether to enrol in research are logistical difficulties.[11 40] Research protocols can be too demanding for participants, especially for those in full-time employment or with caring responsibilities. Patient-focused research can be improved by involving patients early in study design through patient and public involvement groups. In addition to addressing logistical issues, this would highlight important areas of concern that we have found in this study regarding study procedures and allow for considered amendments to protocols; excluding procedures that do not add sufficient value in answering the research question.

### Reassured by healthcare professionals

Pregnant women may be reassured by the stringent requirements placed on researchers by governing bodies. Participants were encouraged that researchers thought the study was of value. Confidence in research staff is important in establishing trust between participants and researchers and may reduce the perception of risk and encourage study enrolment.[11 30] While this may encourage participation in research, it is important that blind faith in research staff is not influencing a participant's decision-making, nor that participants feel pressure to conform to the desires of healthcare staff.[37]

## Altruism

A major influence in favour of enrolling described by pregnant women in this study was altruism. They were encouraged to enrol partly due to potential benefit to their own baby, but also to help other pregnant women, and progress the science-base and care of pregnant women. Altruism driving enrolment has been well established and should be integral to the understanding clinicians have when approaching and enrolling pregnant women in research.[35 41]

### Study strengths, limitations and applications

The study included interviews with 12 pregnant women to assess their views on hypothetical participation in a controlled human infection study. The results of these interviews meaningfully influenced the design and ultimate execution of the Lactamica 9 study, including offering study visits in the participants' homes, limiting study follow-up to 4 months post partum, and collecting umbilical cord blood rather than infant venous blood at birth.[16]

Although the sample size was relatively small, the semi-structured interview format allowed for rich data to be gleaned from each interview and provided detailed themes, which were examined. Interviews were conducted with the group of interest, engaging individuals for whom pregnancy-related research is most important. Discussions were further aided by the capacity participants had gained throughout their pregnancy in weighing decisions and balancing study requirements.

It is noteworthy that interviewees had relatively high educational attainment, and most were ethnically White British. Perceptions appeared consistent between individuals of different educational attainment, but there may be limitations to the generalisability of these results. However, one of the primary aims of this interview study was to inform the eventual design of the Lactamica 9 controlled human infection study, which recruited participants receiving antenatal care at the same large teaching hospital. Thus, although potentially not generalisable beyond the study population, the sample of interviewees may nonetheless be representative of the Lactamica 9 study's recruitment population.

## CONCLUSION

Pregnant participants were keen to be involved in research and were largely driven by altruism. Risk was a major consideration, and their perceptions were influenced by their personal experience. Therapeutic interventions in pregnant women have fallen behind their non-pregnant counterparts following prolonged exclusion from research, which affects treatment options available to this population. Progress has been made in recent years towards the inclusion of pregnant women, but more remains to be done to overcome inequality in research inclusion (see figure 1).

**Contributors** RBD, AAT, RCR, TV and CEJ conceived and designed the study. RBD implemented the study and conducted data analysis. RCR and CEJ provided supervision for the study. TV and CEJ provided supervision of the data interpretation. RBD prepared the manuscript. All authors reviewed, edited and approved the final manuscript. RBD is the author responsible for the overall content and acts as the guarantor.

**Funding** RBD was supported by the University of Southampton National Institute of Health Research Academic Clinical Fellowship Scheme. AAT is supported by the Medical Research Council (Clinical Research Training Fellowship MR/V002015/1).

**Competing interests** None declared.

**Patient and public involvement** Patients and/or the public were involved in the design, or conduct, or reporting, or dissemination plans of this research. Refer to the Methods section for further details.

**Patient consent for publication** Not applicable.

**Ethics approval** This study involves human participants and was approved by Health Research Authority (HRA) London—City & East Research Ethics Committee (Reference: 19/LO/0356). Participants gave informed consent to participate in the study before taking part.

**Provenance and peer review** Not commissioned; externally peer reviewed.

**Data availability statement** The data generated and analysed during the current study are not publicly available because ethical approval for their distribution has not been granted, nor have participants consented to it being made public.

**ORCID iDs**
Robert B Dorey http://orcid.org/0000-0002-5257-7333
Robert C Read http://orcid.org/0000-0002-4297-6728

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
