## [Reviewer comments · BMJ Open]

ARTICLE DETAILS

TITLE (PROVISIONAL)	Qualitative interview study exploring the perspectives of pregnant women on participating in controlled human infection research in the UK
AUTHORS	Dorey, Robert; Theodosiou, Anastasia; Read, Robert; Vandrevalla, T; Jones, Christine

VERSION 1 – REVIEW

REVIEWER	Timothy Doyle Center for Disease Control
REVIEW RETURNED	10-May-2023

GENERAL COMMENTS	A well written paper on an interesting topic. It would be interesting to see how study participants are distributed by trimester of pregnancy, and these results may be worth adding to Table 1. More details on age of participants would also be interesting to know, mean/median, IQR, etc. I would like to know more details about how the sample was chosen, as this impacts the generalizability of the findings. What procedures were used to select these 12 participants? The authors mention (page 9, line 6) that patient and public involvement (PPI) was used to inform design, location, etc., but few details are provided here and nothing about sample selection. The paper could be strengthened if we understood the sampling details better. The authors correctly note the small sample size as a limitation. This should also be reflected by using more caution in statements of conclusion. For example, page 19, line 41, “pregnant women expressed a preference...” One cannot generalize to all pregnant women, based on your small sample, so alternative wording might be something like, “based on our small sample of pregnant women, participants expressed a preference...” The same general concept applies to other portions of the text, (e.g., page 23, line 5, line 27) where the authors appear to overextend their conclusions to pregnant women in general. The small sample size does not support drawing conclusions across such a broad segment of the population, and the language used in conclusions should reflect this. Reference 7 – check accuracy of author list.
--

REVIEWER	Rieke van der Graaf University Medical Center Utrecht
REVIEW RETURNED	05-Jul-2023

GENERAL COMMENTS

I have read the paper with great interest. There is scarce literature on risk perception of pregnant women who participate in research. Moreover, the combination of risk perception and human challenge trials is relatively new.

However, I think a few issues need more attention.

1. It is not clear whether the proposed study in which the pregnant women are hypothetically enrolled is an “interventional study” or specifically a controlled human infection study or challenge study. The terms are used interchangeably in the document. There are several definitions of a CHIM, since this is the UK context I have taken the definition of the UK Parliament. Human challenge studies in the study of infectious diseases - POST (parliament.uk) They define a CHIM as “Human challenge studies are clinical trials in which researchers intentionally expose healthy volunteers to a pathogen (such as a virus, fungus or bacteria). These studies enable researchers to better understand the disease that these pathogens cause. They can also be used to test the effectiveness of drug treatments and vaccines. Human challenge studies have been used since the late 1800s. Since then, human challenge studies have been refined to study infectious disease in controlled settings. People are typically given a small dose of a pathogen through oral ingestion, inhalation, or injection. Medical professionals then closely monitor participants to observe how the body responds and whether treatment is effective in managing or preventing disease.” Could the authors please clarify whether the proposed study is indeed a CHIM? And if so, could the authors give more detail about the proposed controlled environment, the number of participants of the proposed CHIM, the need for a CHIM specifically in pregnant women when the “N. lactamica has been safely administered intranasally to over 400 non-pregnant adults in clinical studies” (ie the social value to be gained)?

2. A bias in this interview study is the high number of White British, highly educated pregnant women who have been interviewed. Has any effort been undertaken to enroll other pregnant women? Why not? Would it be possible to conduct a few more interview studies with other groups of pregnant women? Given the relatively low sample size that would also be good. I can understand that there is saturation in a highly homogeneous group after 12 interviews, but this might change when other subgroups are added. Moreover, informed consent is one of the major ethical issues in CHIMs. The results might change among those who are not high educated.

Minor issues

1. Page 7, line 33 “potentially harmful organism”, line 39 “harmless commensal bacteria”. Which one is correct?

2. Appendix interview guide. Question 3 “We are thinking about doing a study where we would give Neisseria lactamica into the noses of pregnant women.” The authors were interested in a specific reaction, before question 4 is raised they provided the women with more information about the design. However, I’m hesitant about the value of the answer to question 3. Exposing people to Neisseria lactamica without providing any context about the design, among others whether it is a potentially beneficial study or not, will most likely not lead to very informative answers.

3. Although I do see the advantage of interviewing the pregnant

	women before a study is done to inform the eventual design of the study, a limitation of the study is also that it remains hypothetical (recognized in the limitations). It would be really good and interesting to repeat these/similar interviews when the study is actually performed. See also our paper (to which the authors already refer) in which we argued for more research on risk perception with women who are participating in a study instead of hypothetical scenarios (Van der Zande, I.S.E., van der Graaf, R., Oudijk, M.A. et al. A qualitative study on acceptable levels of risk for pregnant women in clinical research. BMC Med Ethics 18, 35 (2017).) Can the authors say anything about plans to repeat this study in the future when the CHIM/interventional study is conducted?
--	---

VERSION 1 – AUTHOR RESPONSE

Reviewer: 1

Dr. Timothy Doyle, Center for Disease Control

Comments to the Author:

A well written paper on an interesting topic.

Dr Doyle, thank you for taking the time to review our manuscript. Please find comments related to your review listed below.

It would be interesting to see how study participants are distributed by trimester of pregnancy, and these results may be worth adding to Table 1. More details on age of participants would also be interesting to know, mean/median, IQR, etc.

Gestation has been added to Table 1 (participants by trimester: 1 in 1st, 6 in 2nd, 5 in 3rd). Data on age was collected into the ranges: 18-24 years, 25-40 years, 41-50 years, 51-60 years, 60 years or older. It was felt that knowing specific ages would not alter data interpretation and so collecting pseudo-anonymised data on age was felt to be preferable with regards to participant confidentiality.

I would like to know more details about how the sample was chosen, as this impacts the generalizability of the findings. What procedures were used to select these 12 participants? The authors mention (page 9, line 6) that patient and public involvement (PPI) was used to inform design, location, etc., but few details are provided here and nothing about sample selection. The paper could be strengthened if we understood the sampling details better.

More clarity of details of PPI has now been provided in the Methods section “Patient and Public Involvement”, as described above. Details about how participants were approached was expanded upon in the first paragraph of the Methods:

“All women attending an antenatal clinic were approached whilst in the waiting room, and their initial interest was determined. Those who expressed interest were given more information about the study, and, if they wished to proceed, informed consent was gained.”

The authors correctly note the small sample size as a limitation. This should also be reflected by using more caution in statements of conclusion. For example, page 19, line 41, “pregnant women expressed a preference...” One cannot generalize to all pregnant women, based on your small sample, so alternative wording might be something like, “based on our small sample of pregnant women, participants expressed a preference...” The same general concept applies to other portions of the text, (e.g., page 23, line 5, line 27) where the authors appear to overextend their conclusions to pregnant women in general. The small sample size

does not support drawing conclusions across such a broad segment of the population, and the language used in conclusions should reflect this.

Language has been adapted throughout the discussion as suggested.

Reference 7 – check accuracy of author list.

Checked and updated: “Adam MP, Polifka JE, Friedman JM m. Evolving knowledge of the teratogenicity of medications in human pregnancy. American Journal of Medical Genetics Part C: Seminars in Medical Genetics. 2011;157(3):175–82.”

Reviewer: 2

Dr. Rieke van der Graaf, University Medical Center Utrecht

Comments to the Author:

I have read the paper with great interest. There is scarce literature on risk perception of pregnant women who participate in research. Moreover, the combination of risk perception and human challenge trials is relatively new.

However, I think a few issues need more attention.

Dr van der Graaf thank you for your input and contribution to this manuscript. Highlighted below are our responses to your feedback.

1. It is not clear whether the proposed study in which the pregnant women are hypothetically enrolled is an “interventional study” or specifically a controlled human infection study or challenge study. The terms are used interchangeably in the document. There are several definitions of a CHIM, since this is the UK context I have taken the definition of the UK Parliament. Human challenge studies in the study of infectious diseases – POST (parliament.uk) They define a CHIM as “Human challenge studies are clinical trials in which researchers intentionally expose healthy volunteers to a pathogen (such as a virus, fungus or bacteria). These studies enable researchers to better understand the disease that these pathogens cause. They can also be used to test the effectiveness of drug treatments and vaccines. Human challenge studies have been used since the late 1800s. Since then, human challenge studies have been refined to study infectious disease in controlled settings. People are typically given a small dose of a pathogen through oral ingestion, inhalation, or injection. Medical professionals then closely monitor participants to observe how the body responds and whether treatment is effective in managing or preventing disease.” Could the authors please clarify whether the proposed study is indeed a CHIM? And if so, could the authors give more detail about the proposed controlled environment, the number of participants of the proposed CHIM, the need for a CHIM specifically in pregnant women when the “N. lactamica has been safely administered intranasally to over 400 non-pregnant adults in clinical studies” (ie the social value to be gained)?

For context, this interview study was undertaken as preliminary work for the world’s first respiratory CHIM study in pregnant women, in which participants were inoculated intranasally with the commensal bacterial *Neisseria lactamica* [Lactamica 9 study; protocol: Theodosiou BMJ Open. 2022;12(5):e056081]. At the time the interviews took place, the CHIM study was still hypothetical; however, the Lactamica 9 study has since been conducted, and the protocol and some results have been published, although most results are still being analysed. The motivations for conducting the Lactamica 9 study are discussed in detail in the published protocol paper, and we have now added some background to this manuscript for clarity. In brief, infant upper respiratory microbes are derived

largely from transmission from the mother during and after birth, and the evolving infant microbiome and immunological profile is known to correlate with downstream health outcomes (including respiratory infection and allergy). The goal of the Lactamica 9 study was to demonstrate whether commensal inoculation in pregnancy results in horizontal transmission to the infant after birth, and whether this could be used as a model to study infant microbiome and immune development.

We conducted this preparatory interview study (the presented manuscript) to: firstly, explore how pregnant women might perceive risk in participating in research, specifically interventional as compared with observational, studies; secondly, to specifically explore pregnant women's perspectives on CHIM, given that such research has not previously been performed in this participant group; and thirdly, to inform design of the Lactamica 9 study. Based on the interviews discussed in this manuscript, the Lactamica 9 protocol was adapted before the CHIM study was carried out. We have updated the manuscript to clarify these points.

2. A bias in this interview study is the high number of White British, highly educated pregnant women who have been interviewed. Has any effort been undertaken to enroll other pregnant women? Why not? Would it be possible to conduct a few more interview studies with other groups of pregnant women? Given the relatively low sample size that would also be good. I can understand that there is saturation in a highly homogeneous group after 12 interviews, but this might change when other subgroups are added. Moreover, informed consent is one of the major ethical issues in CHIMs. The results might change among those who are not high educated.

We agree that this is a limitation of this study, and have updated the 'strengths and limitations' section of the manuscript to reflect your feedback. We did not find significant divergence in perception by educational attainment, and it would be beneficial in future to investigate whether differences in ethnicity affect perceptions of risk. Unfortunately, there is not capacity to undertake further interviews.

Minor issues

3. Page 7, line 33 "potentially harmful organism", line 39 "harmless commensal bacteria". Which one is correct?

Line 33 was intended as a discussion of CHIM in general, including considerations when using organisms such as influenza (or more recently SARS-CoV2). In line 39 we move on to discuss the specific example of what became the Lactamica 9 study, using *Neisseria lactamica* which has been demonstrated to be harmless in the CHIM setting. We have edited the text to reflect this.

4. Appendix interview guide. Question 3 "We are thinking about doing a study where we would give *Neisseria lactamica* into the noses of pregnant women." The authors were interested in a specific reaction, before question 4 is raised they provided the women with more information about the design. However, I'm hesitant about the value of the answer to question 3. Exposing people to *Neisseria lactamica* without providing any context about the design, among others whether it is a potentially beneficial study or not, will most likely not lead to very informative answers.

We agree that there may be limited information that can be gained from an initial response until more information is provided to participants. Our feeling when designing the interview was that decisions of whether to enrol in research may be influenced by a participant's initial emotional response to being approached. Our intention with this question was to explore this, and whether the participants opinion changed as more information was given. There may have been some benefit in this as mentioned in

the Results section 'prior understanding and experiences influence decision-making', and as discussed in the Discussion section 'prior health-related experience affects perception of risk'.

5. Although I do see the advantage of interviewing the pregnant women before a study is done to inform the eventual design of the study, a limitation of the study is also that it remains hypothetical (recognized in the limitations). It would be really good and interesting to repeat these/similar interviews when the study is actually performed. See also our paper (to which the authors already refer) in which we argued for more research on risk perception with women who are participating in a study instead of hypothetical scenarios (Van der Zande, I.S.E., van der Graaf, R., Oudijk, M.A. et al. A qualitative study on acceptable levels of risk for pregnant women in clinical research. BMC Med Ethics 18, 35 (2017).) Can the authors say anything about plans to repeat this study in the future when the CHIM/interventional study is conducted?

We completely agree that there is a paucity of research with pregnant women about both perceptions of research, and also about experiences of those who have engaged in research. Although we do not plan to repeat this study, we were able to collect useful complementary data during the Lactamica 9 study. Participants enrolled in the Lactamica 9 study completed anonymous questionnaires before and after study involvement, to investigate their motivations, concerns and experiences. These results have been published (Bevan et al. A Questionnaire-based Study Exploring Participant Perspectives in a Perinatal Human Challenge Trial. *Pediatr Infect Dis J.* 2023 Jul 31;42(11):935–41.), and we have updated this manuscript to include mention of these data.

VERSION 2 – REVIEW

REVIEWER	Rieke van der Graaf University Medical Center Utrecht
REVIEW RETURNED	08-Dec-2023
GENERAL COMMENTS	The questions that I had have all been adequately addressed, thanks